# Effect of Cathode Physical Properties on the Preparation of Fe_3_Si_0.7_Al_0.3_ Intermetallic Compounds by Molten Salt Electrode Deoxidation

**DOI:** 10.3390/ma15217646

**Published:** 2022-10-31

**Authors:** Hui Li, Yutian Fu, Jinglong Liang, Yu Yang

**Affiliations:** College of Metallurgy and Energy, North China University of Science and Technology, Tangshan 063210, China

**Keywords:** cathode preparation, physical properties, molten salt electrolysis, intermetallic compounds

## Abstract

As a new process, molten salt electrolysis is widely used in the preparation of metal materials by in situ reduction in solid cathodes. Therefore, it is meaningful to study the influence of the physical properties of solid cathodes on electrolysis products. In this paper, mixed oxides of Fe_2_O_3_-Al_2_O_3_-SiO_2_ were selected as raw materials, and their particle size distribution, pore size distribution, specific surface area, and other physical properties were investigated by mechanical ball milling at different times. The CaCl_2_–NaCl molten salt system was selected to electrolyze the sintered cathode solid at 800 °C and a voltage of 3.2 V. The experimental results show that with the prolongation of ball-milling time, the particle size of mixed oxide raw materials gradually decreases, the specific surface area gradually increases, the distribution of micropores increases, and the distribution of mesopores decreases. After sintering at 800 °C for 4 h, the volume and particle size of the solid cathode increased, the impedance value gradually decreased, and the pores first increased and then decreased. The electrolysis results showed that the prolongation of the ball-milling time hindered the electrolysis process.

## 1. Introduction

Iron-based intermetallic compounds have attracted widespread attention because of their excellent properties. Among them, FeSiAl intermetallic compounds have been successfully used as excellent soft magnetic materials in the fields of electronic information and aerospace. Compared with traditional pyrometallurgy and powder metallurgy, molten salt electrolysis has become a hot research topic of many scholars because of its simple technical process, green environmental protection, low temperature, and low consumption process characteristics.

LI et al. [1,2,3] selected the titanium-containing blast furnace slag as a cathode in a molten CaCl_2_ molten salt system, added a certain amount of TiO_2_ and C to it, and finally successfully prepared Ti_5_Si_3_, TiC, and Ti_3_SiC_2_ by electrolysis. On the basis of this method, the preparation of TiAl_3_ alloy and Ti_3_AlC_2_ material is completed by changing the doping material and selecting blast furnace slag containing a certain proportion of Al_2_O_3_ and C as the cathode. Su, X.W. et al. [4] also chose molten CaCl_2_ as the molten salt system, used waste ITO (s-ITO) target as the cathode, and successfully prepared In-Sn alloy by studying the reduction mechanism at different times. Under this process condition, many scholars start from the physical properties of the solid cathode itself. Zhou, Z.R. et al. [5] added a certain amount of NH_4_HCO_3_ to prepare a porous FeTiO_3_ solid cathode and studied the effect of bulk porosity on electrolysis products. Yan et al. [6] showed that the increase in the specific surface area of the solid cathode is beneficial to the diffusion of O^2-^ and the formation of the reaction interface during the electrolysis process, thereby accelerating the electrolytic reduction rate. Li, Z.Q. et al. [7] pointed out that controlling the molding pressure, sintering temperature and specific surface area during the cathode preparation process, and promoting the transfer of metal atoms at the three-phase interface during the electrolysis process are the main problems in improving the electrolysis efficiency.

According to the above research status, in this paper, Fe_2_O_3_-Al_2_O_3_-SiO_2_ mixed oxides were mechanically ball-milled, and the effects of changes in pore distribution, specific surface area, particle size distribution, and porosity and electrical conductivity of the mixed oxides after sintering on the electrolytic preparation of Fe_3_Si_0.7_Al_0.3_ intermetallic compounds were investigated by changing the ball-milling time.

## 2. Materials and Methods

In this study, Fe_2_O_3_, Al_2_O_3_, and SiO_2_ (>99.99%) were selected as raw materials, placed on a planetary ball mill (Shanghai, China) for different times of powdering treatment, and pressed under a pressure of 8 MPa for 2.5 min to form a cylinder (*Φ* = 15 mm, *d* = 1.5 mm) The cylinders were sintered in high-purity Ar (99.99%) at 800 °C for 4 h. The sintered solid cathode is wrapped with a 400-mesh iron mesh cloth and then fixed on a stainless steel rod with an iron wire to prepare a cathode. The polished graphite sheet (>99.99%, 100 × 15 × 3 mm) was repeatedly washed and dried with sufficient deionized water and anhydrous ethanol and then fixed on a stainless steel rod with nickel wire (*Φ* = 2.5 mm) to prepare an anode. The CaCl_2_–NaCl mixed salt (180 g) dried under vacuum at 250 °C for 24 h was placed in a corundum crucible (99.95%) and then placed in a resistance furnace (Luoyang, China). When the temperature reached 800 °C, a 2.8 V constant voltage pre-electrolysis experiment was carried out with the graphite sheet as the anode and the nickel sheet as the cathode to remove the residual moisture and impurity elements in the molten salt [8].

After the pre-electrolysis, the prepared electrodes were put into molten salt, the experimental temperature was 800 °C, the voltage was set to 3.2 V, and the whole process was carried out under an Ar atmosphere. After the electrolysis, the cathode product was removed, soaked in distilled water, cleaned in an ultrasonic cleaner (Kunshan, China), dried in a vacuum drying oven (Shanghai, China) and then removed and sealed.

The particle size range is represented by *W* and is calculated according to Equation (1).
(1)$$W=\frac{\left(M-m_{0}\right)}{M}\times 100\%$$
where *m*_0_ is the mass of the test sieve residue, g; *M* is the mass of the sample g.

The porosity is represented by *P* and is calculated according to Equation (2):(2)$$P=\frac{m_{3}-m_{1}}{m_{2}}\times \;100\%$$
where *P* is the porosity of the sample, *m*_1_ is the dry mass of the sample, g; *m*_3_ is the mass of the water-saturated sample in air, g; *m*_2_ is the weight of the water-saturated sample suspended in water, g.

In the AC impedance detection, the saturated calomel electrode was selected as the reference electrode, the solid cathode sheet was used as the working electrode, the polished graphite sheet (10 × 10 × 2 mm) was used as the auxiliary electrode, and the electrolyte was 0.1 mol/L NaOH solution. The impedance *Z* is calculated according to Equation (3):(3)Z=R+j(XL+XC)
where *Z* is impedance, Ω; *R* is resistance, Ω; *XL* is inductive reactance, Ω; *XC* is capacitive reactance, Ω. The measured AC impedance curve is fitted and analyzed by Zview software (San Francisco, CA, USA).

CHI660E electrochemical workstation (Chinstruments, Shanghai, China) was used for AC Impedance Experiment. Nitrogen adsorption apparatus was selected to detect the pore distribution and specific surface area of the cathode raw material (QUADRASORB evo USA). The phase composition of the cathodic products was analyzed by X-ray diffractometer (XRD, Smartlab SE., Tokyo, Japan).

## 3. Results and Discussion

### 3.1. Cathode Physical Property Research

The N_2_ adsorption–desorption isotherm in Figure 1 is a characteristic curve of type IV mesoporous materials [9]. Figure 2 shows the micropore and mesoporous distribution of mixed oxides. Figure 2a shows that with the extension of ball-milling time, the particle size of the cathode raw material decreases continuously, and the micropore distribution < 0.5 nm gradually increases. After 12 and 14 h of ball milling, there are relatively more large micropores, which is mainly due to the agglomeration phenomenon at small particle size. According to Figure 2b, the mesoporous distribution in the range <20 nm still showed an obvious growth trend, while in the range >30 nm, the mesoporous distribution gradually decreased with the prolonging of the milling time. According to the results in Figure 3, the specific surface area gradually increases with the decrease in particle size under different milling times, and the change range of 14 h is smaller than that of 12 h.

The cathode raw materials after ball milling at different times were sintered, and the particle sizes before and after sintering were sieved. The results are shown in Figure 4. It can be seen from the figure that with the prolongation of the ball-milling time, the overall particle size of the mixed oxide before sintering gradually decreases, and the change in particle size at 14 h is smaller than that at 12 h. After sintering, the oxide cathode particles grow to a certain extent under different ball-milling times [10], and the longer the ball-milling time is, the larger the proportion of larger particles is. This phenomenon is due to the fact that the contact degree between particles is higher after pressing under a certain pressure, and the larger specific surface area is more conducive to the occurrence of nucleation and growth behavior during the sintering process.

Figure 5 is a graph showing the change in the porosity and volume of the solid cathode after sintering with different ball-milling times. It can be seen from the figure that with the extension of the ball-milling time, the volume of the cathode body gradually increases, and this phenomenon is mainly due to the phenomenon of particle growth at a certain sintering temperature. The increase in porosity in the early stage is mainly due to the mutual bonding of particles to form pores. With the increase in specific surface area, the grain growth degree is larger, which eventually leads to the reduction in pores inside the particles. On the basis of the above experimental results and considering the long ball-milling time and energy consumption, the ball-milling time of 8, 10, and 12 h was finally selected for follow-up research.

The AC impedance of the sintered cathode body after different ball-milling times was tested. The curve fitting is shown in Figure 6. In the equivalent circuit, *Rs* is the aqueous solution resistance, Ω; *Rct* is the cathode sheet resistance, Ω; *Wt* is the diffusion impedance, and Ω; CPE1 is a fitting capacitive element. According to Table 1, the impedance value gradually decreases with the increase in ball-milling time. When the ball-milling time is extended to 12 h, the impedance value changes obviously. This is mainly due to the existence of more dislocation defects in the particles caused by the long-time ball milling, which reduces the transmission resistance of electrons and reduces the impedance value [11].

### 3.2. Influence of Physical Properties on Electrolysis Products

To investigate the effects of different physical properties on the electrolysis products of Fe_2_O_3_-Al_2_O_3_-SiO_2_ cathode, electrode deoxidation experiments were carried out at 800 °C with a constant voltage of 3.2 V and a reaction time of 12 h. The XRD results of the electrolysis products are shown in Figure 7.

It can be seen from Figure 7 that after ball milling for 8 h, the electrolysis products are Fe_3_Si_0.7_Al_0.3_ and CaCO_3_, while Si_5_C_3_ appears in the cathode reduction product of ball milling for 10 h. The formation of CaCO_3_ is due to the fact that the CO_2_ generated by the anode further forms CO_3_^2-^ with O^2-^ in the molten salt, and CO_3_^2–^ forms CaCO_3_ with Ca^2+^ near the cathode [12]. The reaction is shown in Equations (4)–(6). At the same time, the electrolysis reaction of CO_3_^2-^ also occurs in the molten salt [12]. As the particle size of the cathode decreases, the larger specific surface area makes the formed elemental C and the Si in the cathode form Si_5_C_3_. The reaction is shown in Equations (7) and (8). The C-cycle schematic is shown in Figure 8.
(4)$${\;C+2O}^{2-}{=\mathrm{CO}}_{2}(g)+4e^{-\;}$$
(5)$${\mathrm{CO}}_{2}{+O}^{2-}{=\mathrm{CO}}_{3}{}^{2-\;}$$
(6)$${\mathrm{CO}}_{3}^{2-}{+\mathrm{Ca}}^{2+}{=\mathrm{CaCO}}_{3}$$
(7)$${\mathrm{CO}}_{3}^{2-}{+4e}^{-}{=C+3O}^{2-\;}$$
(8)$${3C+5\mathrm{Si}=\mathrm{Si}}_{5}C_{3}$$

With the further extension of the ball-milling time, Fe_3_Si_0.5_Al_0.5_, Si_5_C_3_, Fe_3_Si, and CaSiO_3_ appeared in the reduction products. Among them, CaSiO_3_ is formed from the formation of SiO_2_ and CaO in the molten salt, and the reaction is shown in Equation (9). Combined with HSC Chemistry6.0 thermodynamic calculation software (Outokumpu, Finland) and through calculation, it can be known that the theoretical decomposition voltages of SiO_2_ and CaSiO_3_ are –1.86 V and –2.09 V, respectively. Higher thermodynamic conditions will hinder the progress of the electrolysis process so that there is not enough time for the diffusion of metal elements. According to the above experimental results, it can be seen that the extension of the ball-milling time has hindered the electrolysis process.
(9)$${\mathrm{CaO}+\mathrm{SiO}}_{2}{=\mathrm{CaSiO}}_{3}$$

Figure 9a shows the *I*–*t* curves obtained by electrolysis under different ball-milling time conditions. The higher current at the beginning of the curve is mainly due to the charging process of the electric double layer. As the electrolysis reaction progresses from the outside to the inside, the reaction interface decreases, and the current gradually decreases. The comparison of physical properties shows that the porosity of the solid cathode prepared after 8 h ball milling is smaller than the other two, the particle size and resistance are relatively larger, the curve of 4–12 h tends to be flat, the current is basically stable, and the reaction speed is relatively slow. After 10 h of ball milling, the particle size and resistance of the cathode decreased, which was beneficial to the electrolysis process. It can be seen from the graph that the slope of the curve in the early stage is larger, indicating that the electrolysis reaction speed is faster. In the later stage of electrolysis, there is more dissolved CaO in the molten salt, which makes the background current larger [13]. After 12 h of ball milling, the particle size and resistance of the cathode were further reduced, and the overall *I*–*t* curve fluctuated greatly, indicating that there were more side reactions [14]. At the same time, CaSiO_3_ still exists in the final product, indicating that this physical condition inhibits the progress of the electrolysis process so that Si is not completely reduced.

Comparing the morphology of anode graphite sheet after electrolysis in Figure 9b, it can be seen that with the extension of ball-milling time, smaller particle size, impedance value, and larger specific surface area are conducive to the removal of O^2-^ during electrolysis and the oxidation of O^2-^ at the anode. The above phenomena make the surface of the graphite sheet more porous. However, the higher O^2-^ concentration will cause the phenomenon of concentration polarization near the cathode, which will lead to the deterioration of the kinetic conditions of the entire electrolysis process, resulting in a slow electrolysis process.

## 4. Conclusions

By studying the physical properties of Fe_2_O_3_-Al_2_O_3_-SiO_2_ mixed oxides before and after sintering by different ball-milling times, combined with the XRD results of molten salt electrolysis, the following conclusions were obtained:

(1) With the increase in ball-milling time, the particle size of Fe_2_O_3_-Al_2_O_3_-SiO_2_ mixed oxide before sintering gradually decreases, the distribution of micropores increases, and the distribution of mesopores decreases. The specific surface area increases with the decrease in particle size, and the sintered cathode particles have obvious nucleation and growth phenomenon, accompanied by the generation of pores;

(2) With the extension of the ball-milling time, the impedance value of the solid cathode after sintering gradually decreases. When it reaches 12 h, the change is more obvious. The phenomenon is mainly that the particles themselves have more dislocation defects over a longer time, which further reduces the electron transport resistance;

(3) After 3.2 V constant voltage electrolysis for 12 h, the CaCO_3_ in the products was attributed to the C cycle caused by the graphite anode, and with the extension of the ball-milling time, the smaller particle size made Si_5_C_3_ appear in the electrolysis products. When the ball-milling time reaches 12 h, the larger specific surface area and the smaller impedance value are conducive to the removal of O^2-^ and migration to the anode discharge. However, because of the poor dynamic conditions such as concentration polarization, the *I*–*t* curve fluctuates obviously. The formation of CaSiO_3_ further impedes the electrolysis process.

## Figures and Tables

**Figure 1 materials-15-07646-f001:**
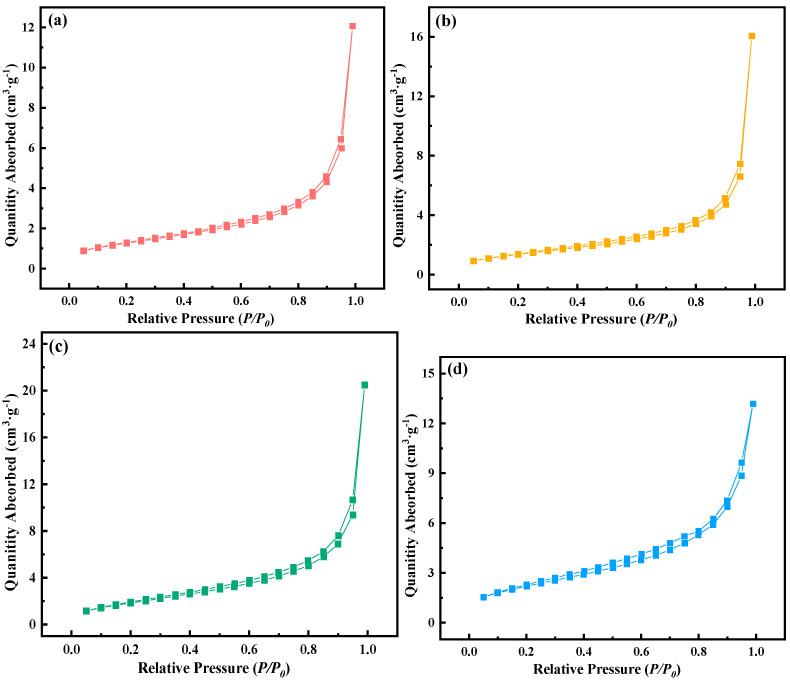
Isothermal curves of mixed oxides after different ball-milling times. ((**a**) 8 h, (**b**) 10 h, (**c**) 12 h, (**d**) 14 h).

**Figure 2 materials-15-07646-f002:**
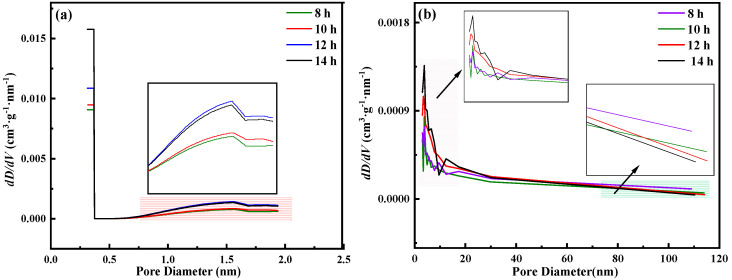
Distribution of micropores (**a**) and mesopores (**b**) in ternary oxides under different ball-milling times.

**Figure 3 materials-15-07646-f003:**
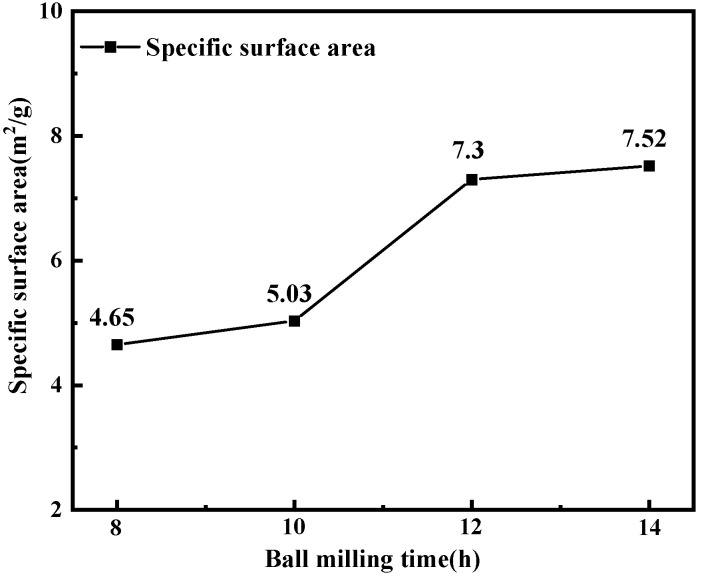
Changes in specific surface area of mixed oxides after ball milling at different times.

**Figure 4 materials-15-07646-f004:**
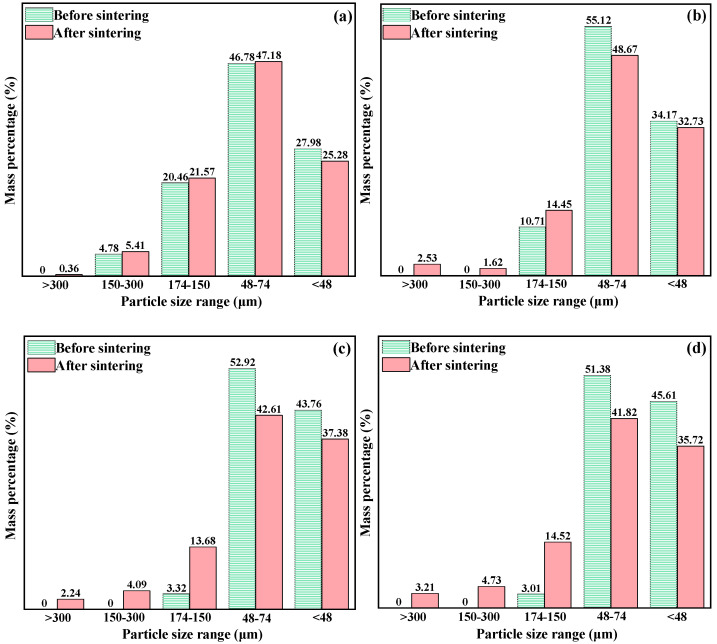
Cathode particle size distribution of mixed oxides after different ball-milling times. ((**a**) 8 h, (**b**) 10 h, (**c**) 12 h, (**d**) 14 h).

**Figure 5 materials-15-07646-f005:**
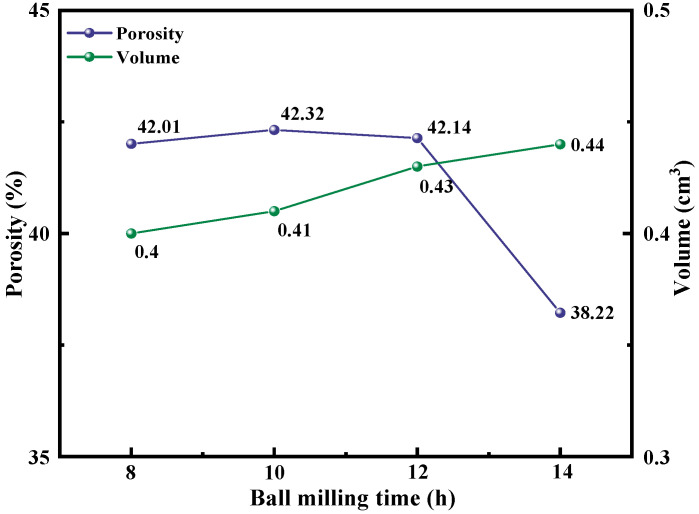
Changes of porosity and volume of mixed oxides after sintering at 800 °C for 4 h.

**Figure 6 materials-15-07646-f006:**
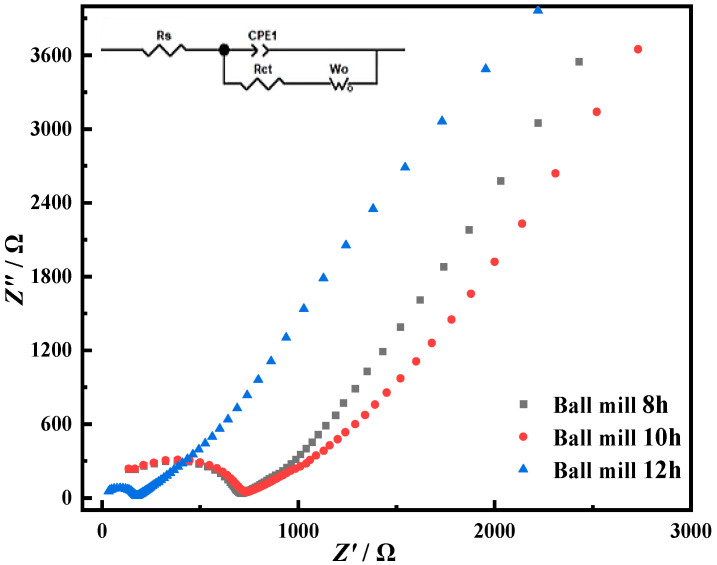
AC impedance spectrum of mixed oxide after sintering.

**Figure 7 materials-15-07646-f007:**
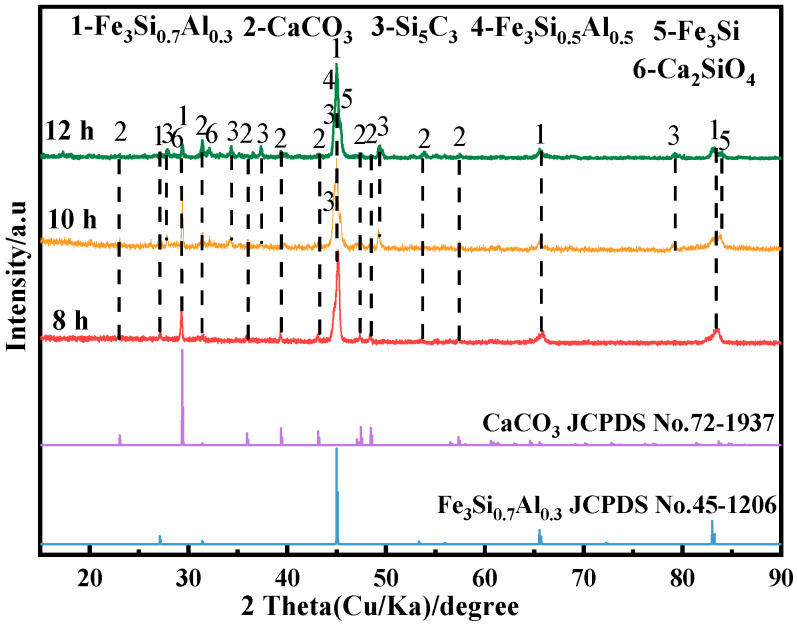
XRD patterns of the solid cathode electrolyzed in CaCl_2_–NaCl molten salt at 800 °C at a constant voltage of 3.2 V for 12 h.

**Figure 8 materials-15-07646-f008:**
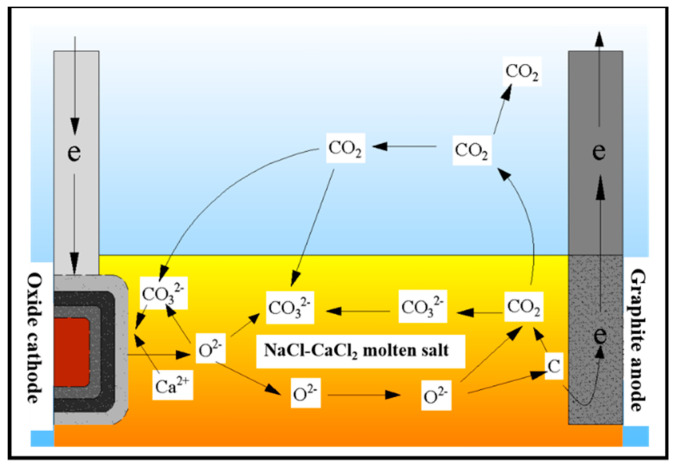
Schematic diagram of C cycle.

**Figure 9 materials-15-07646-f009:**
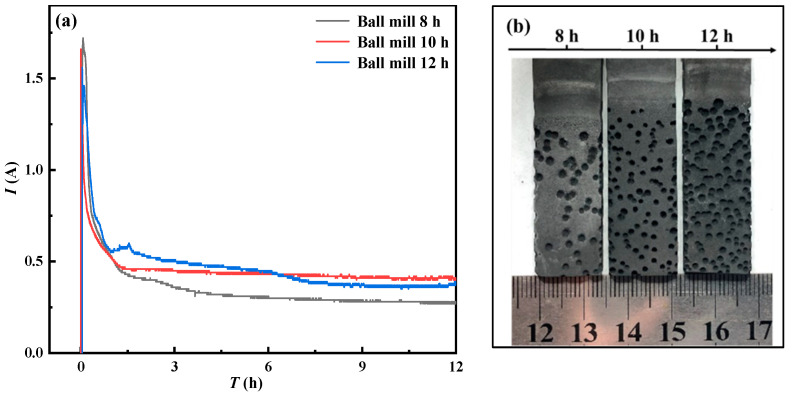
*I*–*t* curve of mixed oxide cathode after electrolysis at 3.2 V constant voltage in 800 °C CaCl_2_–NaCl molten salt for 12 h (**a**) and the graph of graphite sheet after electrolysis (**b**).

**Table 1 materials-15-07646-t001:** Impedance values of mixed oxides after sintering.

Ball-Milling Time/h	*Rct*/Ω
8	598.6
10	560.8
12	162.6

## Data Availability

Data sharing is not applicable to this article.

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
