# Peer review of "Effect of Cathode Physical Properties on the Preparation of Fe3Si0.7Al0.3 Intermetallic Compounds by Molten Salt Electrode Deoxidation"

_materials, 2022, doi:10.3390/ma15217646_

Round 1

Reviewer 1 Report

This work is a development of the FFS Cambridge method of obtaining metals and alloys from their oxides by electrolysis.

Author Response

Dear  Reviewers, according to your comments, the response is as follows.

Reviewer 2 Report

The scientific questions solved in this work are interesting and useful, but the format of presenting the results leaves much to be desired. The experimental part raises many questions, because perhaps it is not very clearly written. When discussing the results, there is a lot of understatement.

 Comments:

1) Chapter 2 (first paragraph), lines 54-57. Synthesis conditions:

The composition of the oxide mixture is not given, and the annealing temperature is not justified. Did the authors change the composition of the mixture?

2) Chapter 2 (first paragraph), ls. 63-66. Purification electrolysis:

Why were such conditions of purification electrolysis chosen? What are the electrolyte requirements? How efficient is electrolysis? that is, the degree of purity of the resulting electrolyte? Perhaps the purification electrolysis is not necessary at all?

3) Chapter 2. ls. 67-69. Electrolysis:

Why were such conditions of electrolysis chosen? (T, V ?)

4) The Equation (line 79) calculates P. Is it total porosity or macroporosity? Further in the text, the authors use the values of microporosity and mesoporosity. How did they get them?

5) l.84. The phrase “… the solid cathode sheet was used as the working electrode” - missing cathode material. Were the cathode sheets made of the sintering cylinders?

6) Impedance (ls. 83-89). What frequency range was applied?

7) Ls. 104-106. Please clarify the phrase. “The N2 adsorption-desorption isotherm in Figure 1 clearly shows a type IV isotherm with an H4 hysteresis loop, which belongs to a characteristic curve of mesoporous materials [9]”.  In which of the figures (ls.97-99) the hysteresis loop appears is not at all obvious. In addition, 4 figures called Figure 1 are not additionally marked in any way.

8) Figure 2. How were the values D and V in Figure 2 obtained?

8) Ls. 107-112.

The statements given are not at all obvious from Figure 2. It seems that the results obtained at 8, 10, 12 and 14 hours are almost the same. Otherwise, please explain in more detail.

9) L.150. “…Rct is the anode sheet resistance, Ω;” Is it correct?

10) General remark.

The chapter should not start with pictures. It is much easier to understand the text when illustrations and tables are given after a mention in the text and some preamble.

11) Many unclear phrases, slang expressions that are unacceptable in a scientific article.

Just for example (this is not all):

l.12 – “…other physical properties of Fe2O3, Al2O3, SiO2 mixed oxides were studied by mechanical ball milling…”

l.48 – “…the Fe2O3, Al2O3, SiO2 mixed oxides were mechanically ball-milled in this paper.”

l.71 – “After the electrolysis, the cathode product after electrolysis in the iron mesh…”

l.72 – “…soaked in distilled water and the molten salt in the sample was removed..”

Author Response

(The authors gave the same response as above.)

Reviewer 3 Report

In this paper, the particle size distribution, pore distribution, specific surface area and other physical properties of Fe2O3, Al2O3, SiO2 mixed oxides were studied by mechanical ball milling for different times. The CaCl2 - NaCl molten salt system was selected to electrolyze the sintered cathode solid at 800 °C and a voltage of 13 3.2 V. The experimental results show that with the prolongation of ball milling time, the particle size of mixed oxide raw materials gradually decreases, the specific surface area gradually increases, the distribution of micropores increases, and the distribution of mesopores decreases. After sintering at 800 °C for 4 h, the volume and particle size of the solid cathode increased, the impedance value gradually decreased, and the pores first increased and then decreased. The electrolysis results showed that the prolongation of the ball milling time hindered the electrolysis process. This research study seems interesting and covers timely subject in the field. The quality of manuscript is up to standard level and methodology and results are represented precisely. This manuscript could be accepted after incorporation of comprehensive proofread throughout the manuscript to rectify typo/grammatical errors. 

Author Response

(The authors gave the same response as above.)

Round 2

Reviewer 2 Report

I thank the authors for taking my comments into account. There are no more comments.